# SBA-15 with Crystalline Walls Produced via Thermal Treatment with the Alkali and Alkali Earth Metal Ions

**DOI:** 10.3390/ma14185270

**Published:** 2021-09-13

**Authors:** Sung Soo Park, Sang-Wook Chu, Liyi Shi, Shuai Yuan, Chang-Sik Ha

**Affiliations:** 1Department of Polymer Science and Engineering, School of Chemical Engineering, Pusan National University, Busan 46241, Korea; nanopss@pusan.ac.kr (S.S.P.); ipris2000@naver.com (S.-W.C.); 2Research Center of Nanoscience and Nanotechnology, Shanghai University, Shanghai 200444, China; shiliyi@shu.edu.cn (L.S.); s.yuan@shu.edu.cn (S.Y.); 3Emerging Industries Institute, Shanghai University, Jiaxing 314006, China

**Keywords:** crystalline SiO_2_ structure, mesoporous silica, rearrangement of silica matrix, alkali metal ions, thermal treatment

## Abstract

Crystalline walled SBA-15 with large pore size were prepared using alkali and alkali earth metal ions (Na^+^, Li^+^, K^+^ and Ca^2+^). For this work, the ratios of alkali metal ions (Si/metal ion) ranged from 2.1 to 80, while the temperatures tested ranged from 500 to 700 °C. The SBA-15 prepared with Si/Na^+^ ratios ranging from 2.1 to 40 at 700 °C exhibited both cristobalite and quartz SiO_2_ structures in pore walls. When the Na^+^ amount increased (i.e., Si/Na increased from 80 to 40), the pore size was increased remarkably but the surface area and pore volume of the metal ion-based SBA-15 were decreased. When the SBA-15 prepared with Li^+^, K^+^ and Ca^2+^ ions (Si/metal ion = 40) was thermally treated at 700 °C, the crystalline SiO_2_ of quartz structure with large pore diameter (i.e., 802.5 Å) was observed for Ca^+2^ ion-based SBA-15, while no crystalline SiO_2_ structures were observed in pore walls for both the K^+^ and Li^+^ ions treated SBA-15. The crystalline SiO_2_ structures may be formed by the rearrangement of silica matrix when alkali or alkali earth metal ions are inserted into silica matrix at elevated temperature.

## 1. Introduction

Mesoporous silica like MCM- and SBA-series is a very interesting material in the field of nanotechnology [1,2] and has been prepared by the templating method using surfactants and/or polymers as templates, have tunable pore sizes and mesostructures and high surface areas [1,2,3,4,5,6]. Mesoporous silicas can be applied in catalysis [7,8,9,10], sorption [11,12,13,14,15,16,17,18], separations [19,20,21,22,23,24], drug delivery [25,26,27,28,29,30], optics [31,32], sensing [33,34,35], devices [36] and imaging [37,38].

Mesoporous silica materials with metal (i.e., Al, Ga, In, Ti, Zr and Zn, etc.) oxide incorporated frameworks have greatly expanded and advanced their applications such as catalysis and magnetics [6]. The incorporated nanoparticles can provide framework with crystalline structures. Meanwhile, periodic mesoporous organosilicas (PMOs) with crystal-like layered structures in frameworks were synthesized using bridged organosilica precursors including aromatic groups such as benzene and biphenyl groups as silica sources [26,39,40]. Mesoporous materials with inorganic crystalline frameworks were also synthesized with zeolite, which was well-known as microporous material with well-defined structure [41,42,43]. The resulting zeolite materials with hierarchical pore size distributions exhibit significantly improved catalytic properties for various reactions with respect to microporous zeolite materials [41,42,43].

Crystalline SiO_2_ materials may show enhanced chemical and thermal stability in comparison with amorphous SiO_2_ materials [44,45]. Crystalline silicas were prepared from amorphous silica using carbon [46], water [47], noble metals [48,49] and alkali metal ions [45,50,51,52,53,54] as catalyst in previous literatures. Amorphous silica might be also crystallized at high temperature and high pressure. Santoro et al. [55] prepared crystalline CO_2_-SiO_2_ solid solution by reacting CO_2_ with silica at high pressure and temperature. At high pressure, a phase transformation was observed by Tsuchida and Yagi [56] from cristobalite and quartz to stishovite and another phase. At high pressure, a new polymorphic phase transition of coesite to post-stishovite was discovered even at room temperature by Hu et al. [57]. Huang et al. [58] also reported a transformation from cristobalite to stishovite at high pressure.

The amorphous SiO_2_ microtubes can be crystallized when they are catalyzed by LiNb(OC_3_H_7_)_6_ [45]. Silica glass can be crystallized using sodium chloride as catalyst [51]. Venezia et al. [52] found that alkali ions can affect on the transition of amorphous to crystalline phase of silica powders. Shinohara et al. [53] observed cristobalite and tridymite crystalline structures when they are prepared in rice husk ash containing 93% SiO_2_ and 2–3% K_2_O by thermal treatment above 900 °C. Drisko et al. [59] synthesized hollow silica with crystalline wall using hollow mesoporous silica as source of silica material and Mg^2+^, Ca^2+^, Sr^2+^ and Ba^2+^ as metal ion sources.

We synthesized carbon/mesoporous silica hybrid films with crystalline silica wall by thermally treating the ethylene-bridged mesoporous organosilica in the presence of alkali metal ion (Na^+^) in the previous work [44].

In this work, we synthesized SBA-15 with crystalline walls and expanded mesopore sizes by thermally treating SBA-15 with alkali and alkali earth metal ions (Li^+^, Na^+^, K^+^ and Ca^2+^) as catalyst.

## 2. Materials and Methods

### 2.1. Materials

Poly(ethylene glycol)-*b*-poly(propylene glycol)-*b*-poly(ethylene glycol) (Pluronic P123) (Mw = 5800), tetraethylorthosilicate (TEOS, 98%), NaCl (≥99.5%), KCl (≥99.0%), LiCl (≥99.0%) and CaCl_2_ (≥97%) were obtained from Sigma-Aldrich Chemicals. Hydrochloric acid (HCl, 35%) was obtained from Junsei Chemicals (Tokyo, Japan). All chemicals were used as received. Distilled and deionized (DI) water was used throughout all the syntheses.

### 2.2. Synthesis of SBA-15

SBA-15 was synthesized by the similar method as reported elsewhere [2,60]; in a typical synthesis, 4 g of the triblock copolymer was dissolved in 125 g of DI water and 20 g of con. HCl was added with stirring at 35 °C for 1 h. 8.5 g of TEOS was added to the solution, followed by stirring the mixture at 35 °C for 24 h. Then, it was kept at 100 °C for 24 h in a Teflon bottle. The crystallized product was filtered, washed with water, dried, followed by calcinating at 550 °C for 5 h in air to remove occluded template.

### 2.3. Synthesis of SBA-15 with Crystalline Walls

A total of 5 mL of sodium chloride aqueous solutions were added to 0.5 g of SBA-15. Sodium chloride aqueous solutions were prepared with different concentrations (Si/Na = 2.1–80). The slurries were stirred at 25 °C for 1 h. After that, water was removed by evaporation method under vacuum at 80 °C and then the samples were heated in the range of 500–700 °C for 3 h in air. Mesoporous silica with crystalline walls were also synthesized using other metal ions (K^+^, Li^+^ and Ca^2+^) aqueous solutions. The other process was the same as that for mesoporous silica with crystalline walls synthesized using sodium chloride solution. The materials synthesized were called as SBA15-A-C-T, where A represents metal ion species, C represents the ratio of Si/A, T is the thermal treatment temperature.

### 2.4. Characterization

Small-angle X-ray scattering (SAXS) patterns were obtained using the synchrotron X-ray source of the Pohang Accelerator Laboratory (PAL) (Pohang University of Science and Technology, Pohang, Korea) with Co-Kα (λ = 1.608 Å) radiation. X-ray diffraction (XRD) patterns were obtained using an X-ray diffractometer (Rigaku, Tokyo, Japan) (Rigaku Miniflex, Cu-Kα (λ = 1.5418 Å)). Morphology was observed using scanning electron microscopy (SEM, JEOL 6400) (Rigaku, Tokyo, Japan) (acceleration voltage; 20 kV) and transmission electron microscopy (TEM, H-7600) (Hitachi, Tokyo, Japan) with accelerating voltage of 100 kV). The N_2_ adsorption/desorption isotherms were obtained at 77 K using Micromeritics ASAP 2010. Before testing, samples were degassed at 423 K for 12 h. The specific surface area and pore size distribution were analyzed using the Brunauer–Emmet–Teller (BET) method and the Barrett–Joyner–Halenda (BJH) method, respectively.

## 3. Results

Figure 1 illustrates the synthesis of SBA-15 (I) and SBA-15 with crystalline pore walls (II). SBA-15 with 2D-hexagonal mesostructure was synthesized via self-assembly and hydrothermal reaction at 100 °C for 24 h under acidic condition using TEOS as a silica source and using P123 as a structure-directing agent (Figure 1(I)). The template in mesostructured silica was removed by calcination at 550 °C for 5 h in air. Mesoporous silica with crystalline pore walls was prepared via high thermal treatment (500–700 °C) process with various alkali (Na^+^, Li^+^, K^+^) or alkali earth metal (Ca^2+^) ions (Figure 1(II)). The mesoporous silica walls produced crystalline silica species of cristobalite and quartz. The detailed experimental process was described in the experimental section.

Figure 1A shows SAXS patterns of (a) SBA-15, (b) SBA15-Na-80-700, (c) SBA15-Na-60-700, (d) SBA15-Na-50-700, (e) SBA15-Na-40-700, (f) SBA15-Na-30-700, (g) SBA15-Na-20-700 and (h) SBA15-Na-2.1-700. Inset shows the magnified SAXS patterns in the q = 0.10–0.20 ranges for samples. The SAXS pattern of mesoporous silica, SBA-15 (Figure 1A(a)) showed three scattering peaks of the 100, 110 and 200 reflections of the hexagonal symmetry lattice of the SBA-15 [2,60]. Though the intensities of the scattering peaks were decreased after thermal treatment at 700 °C when the Si/Na ratio was decreased to 50 (Figure 1A(d)), well-resolved three scattering peaks (100, 110, 200) of the hexagonal symmetry were still observed. On the other hand, the characteristic peaks in SAXS patterns disappeared after thermal treatment at 700 °C when the Si/Na ratios were above 40 (Figure 1A(e)).

Figure 1B shows XRD patterns of (a) SBA-15, (b) SBA15-Na-80-700, (c) SBA15-Na-60-700, (d) SBA15-Na-50-700, (e) SBA15-Na-40-700, (f) SBA15-Na-30-700, (g) SBA15-Na-20-700 and (h) SBA15-Na-2.1-700 in the range of 2θ = 10–80°. SBA15 (Figure 1B(a)) and samples treated with Si/Na ratios below 50 (Figure 1B(b–d)) showed broad peaks about 2θ = 22°. Meanwhile, when the Na^+^ amount was increased to Si/Na = 40, a weak peak appeared at 2θ = 21.9° indicating the production of cristobalite [50,51,52,53,54]. When the Na^+^ amount was increased up to Si/Na = 2.1 (Figure 1B(h)), the intensity of cristobalite peaks that appeared at 2θ = 21.9, 28.4, 31.6, 40.8, 46.8 and 48.4° in XRD patterns increased due to the increased amount of cristobalite species. At this time, another different type of crystalline silica, quartz, was also produced with Na_2_O generated from oxidation of NaCl after thermal treatment at 700 °C in air [50,51,52,53,54].

Figure 2 displays SAXS (A) and XRD (B) patterns of (a) SBA15-Na-40-500, (b) SBA15-Na-40-600 and (c) SBA15-Na-40-650 after thermal treatment with different temperatures (500–650 °C) using Si/Na = 40. Inset shows the magnified SAXS patterns in the range of q = 0.10–0.20 for samples. After thermal treatment at low temperatures of 500 and 600 °C (Figure 2B(a,b)), crystalline silicas were not produced in the samples while the peaks due to NaCl were appeared at 2θ = 27.3, 31.6, 45.3, 53.7, 56.3, 66.1 and 75.2° (ICDD card No. 05-0628). As shown in Figure 2A(b), The ordering of mesoschannels in the sample was decreased remarkably after thermal treatment at 600 °C (SBA15-Na-40-600). After thermal treatment at 650 °C (Figure 1A(c),B(c)), cristobalite was produced with a small peak at 2θ = 21.9°, while the mesostructure of the sample was collapsed.

Figure 3 shows SEM images of (a) SBA-15, (b) SBA15-Na-80-700, (c) SBA15-Na-60-700, (d) SBA15-Na-50-700, (e) SBA15-Na-40-700, (f) SBA15-Na-30-700, (g) SBA15-Na-20-700 and (h) SBA15-Na-2.1-700. SBA-15 (Figure 3a) and the thermal treated samples at 700 °C with Si/Na ratios below 50 (Figure 3b–d)) have short-rod morphologies. When the Si/Na ratios were above 40, the particles of large pores were produced and the morphology of particles was changed to spherical shapes. These results can be also observed in TEM images (Figure 4). The ordered 2D-hexagonal mesostructure of SBA-15 was collapsed with the increasing amount of alkali metal ion (Na^+^) up to Si/Na = 2.1 (Figure 4h), while the size of mesopores became larger.

Figure 5 shows SEM (a–c) and TEM images (d–f) of (a,d) SBA15-Na-40-500, (b,e) SBA15-Na-40-600 and (c,f) SBA15-Na-40-650 after thermal treatment with different temperatures (500–650 °C) using Si/Na = 40. As displayed in Figure 5a–c, short-rod morphologies were observed for all samples. In particular, SBA15-Na-40-650 has large pores after thermal treatment at 650 °C (Figure 5c). As illustrated in Figure 5d,e, the samples synthesized below 600 °C have the ordered mesostructures. On the other hand, the sample synthesized at 650 °C has the collapsed mesostructure with the expanded mesopores.

Figure 6 shows the nitrogen adsorption/desorption isotherm curves (A) and the pore size distributions (B) of (a) SBA-15, (b) SBA15-Na-80-700, (c) SBA15-Na-60-700, (d) SBA15-Na-50-700, (e) SBA15-Na-40-700, (f) SBA15-Na-30-700, (g) SBA15-Na-20-700 and (h) SBA15-Na-2.1-700. Isotherm curves of SBA15 (Figure 6A(a)) and the samples synthesized with Si/Na ratios below 40 (Figure 6A(b–d)) displayed type IV patterns as for the IUPAC classified common mesoporous materials with cylindrical mesopores [44]. In addition, the four samples (Figure 6B(a–d)) have narrow pore size distributions. Physico-chemical properties of the samples synthesized at 700 °C with different amount of alkali metal ion (Na^+^) was listed in Table 1. When the alkali metal ion (Na^+^) amount was increased to Si/Na = 2.1, BET surface area and pore volume were decreased from 713 to 11 m^2^·g^−1^ and from 0.92 to 0.03 cm^3^·g^−1^, respectively. On the other hand, pore diameter was increased from 66.7 to 1779.9 Å.

Figure 7 shows the nitrogen sorption isotherms (A) and pore size distributions (B) of (a) SBA15-Na-40-500, (b) SBA15-Na-40-600 and (c) SBA15-Na-40-650 after thermal treatment with different temperatures (500–650 °C) using Si/Na = 40. Except SBA-Na-40-650, both samples (SBA15-Na-40-500 and SBA15-Na-40-600) displayed type IV patterns, as explained above, with narrow pore size distributions [44]. Physico-chemical properties of the samples synthesized with different temperatures (500–650 °C) using Si/Na = 40 was listed in Table 2. With the increase of temperature from 500 to 600 °C, BET surface area and pore volume were decreased from 477 to 25 m^2^·g^−1^ and from 0.73 to 0.04 cm^3^·g^−1^, respectively. The sample synthesized at 650 °C (SBA15-Na-40-650) has large pores with dual pore size (118.9 and 366.3 Å) due to the expanded mesopores after thermal treatment under alkali metal ions (Na^+^).

Figure 8A shows XRD patterns in the range of 2θ = 1.2–6° of (a) SBA15-Li-40-700, (b) SBA15-K-40-700 and (c) SBA15-Ca-40-700 after thermal treatment at 700 °C using different metal ions (Li^+^, K^+^ and Ca^2+^) with Si/metal ion = 40. Figure 8B shows the XRD patterns in the range of 2θ = 10–80° for samples. SBA15-Li-40-700 and SBA15-K-40-700 synthesized after thermal treatment at 700 °C using Li^+^ and K^+^ as metal ion sources retained ordered mesostructures with two peaks of the 110 and 200 reflections of the hexagonal mesoporous materials [2,60], even though SBA15-K-40-700 has low intensity of the two peaks of the 110 and 200 reflections (Figure 8A(b)).

On the other hand, the peaks (110 and 200) of SBA15-Ca-40-700 were disappeared (Figure 8A(c)). The result indicates that the mesostructure of SBA15-Ca-40-700 was collapsed after thermal treatment in the presence of Ca^2+^ ions. Interestingly, SBA15-Ca-40-700 synthesized using Ca^2+^ as alkali earth metal ion source produced quartz species mainly with the peaks at 2θ = 20.8, 26.6, 36.4, 39.5, 42.4, 45.8, 50.1, 54.8, 59.9 and 68.1° as shown in the XRD result of Figure 8B(c) [59].

However, SBA15-Li-40-700 prepared with Li^+^ (ionic radius of 0.76 Å) (Figure 8(a)) and SBA15-K-40-700 prepared with K^+^ (ionic radius of 1.38 Å) (Figure 8(b)) did not produce crystalline silica species under the condition in this work.

Figure 9 shows SEM (a–c) and TEM (d–f) images of (a,d) SBA15-Li-40-700, (b,e) SBA15-K-40-700 and (c,f) SBA15-Ca-40-700 after thermal treatment at 700 °C using different metal ions (Li^+^, K^+^ and Ca^2+^) with Si/metal ion = 40. The particles have short-rod and spherical morphologies as shown in SEM images (Figure 9a–c). SBA15-Ca-40-700 showed expanded pores with large size by the rearrangement of silica matrix by incorporating Ca^2+^ ions into silica matrix inducing crystallization with thermal treatment at high temperature (Figure 9f). Pore size, surface area and pore volume of SBA15-Ca-40-700 were as 802.5 Å, 35 m^2^ g^−1^ and 0.27 cm^3^ g^−1^, respectively, from N_2_ sorption result (Figure 10, Table 3). On the other hand, SBA15-Li-40-700 and SBA15-K-40-700 retained 2D-hexagonal mesostructure as shown Figure 9d,e [60]. Table 3 summarizes pore size, surface area and pore volume for both samples.

## 4. Discussion

In this work, we synthesized mesoporous silica with crystalline walls in the presence of alkali and alkali earth metal ions (Li^+^, Na^+^, K^+^, Ca^2+^) via thermal treatment. The crystalline walled SBA-15 has also large mesopore size when silica matrix was rearranged in pore walls. We characterized the crystalline wall structures of SBA-15 prepared with different thermal treatment conditions and different ratios of Si to alkali or alkali earth metal ions (Na^+^, Li^+^, K^+^ and Ca^2+^) by XRD patterns, since the XRD patterns clearly evidenced different crystalline structures of the walls, though other sophisticated techniques including electron diffraction on a transmission electron microscopy may provide more accurate nature of the crystalline wall structures.

Figure 1A shows SAXS patterns of samples synthesized with various content (Si/metal ion = ∞–2.1) of Na^+^ at 700 °C. Three peaks (100, 110, 200) in SAXS patterns were retained after thermal treatment at 700 °C with the content of Na^+^ in the range of Si/Na = ∞~30. In particular, the well-reserved 100 scattering peak indicates the two-dimensional (2D) hexagonal mesostructure. On the other hand, the characteristic peaks in SAXS patterns disappeared with the content of Na^+^ in the range of Si/Na = 40–2.1. The result indicates that the arrangement of mesochannels was collapsed at high temperature of 700 °C with the high amount of Na^+^. The collapse is due to the rearrangement of the silica matrix by Na^+^ inserted into silica.

As shown in XRD patterns of Figure 1B, when the Na^+^ amount was increased to Si/Na = 40, interestingly, a weak peak appeared at 2θ = 21.9° (Figure 1B(e)). The peak can be attributed to the cristobalite, one of silicas with crystalline structure [50,51,52,53,54]. When the Na^+^ amount was increased up to Si/Na = 2.1 (Figure 1B(h)), the intensity of cristobalite peaks increased due to the increased amount of cristobalite species.

It was reported that at low pressure such as atmospheric pressure, quartz, tridymite and cristobalite are stable up to 870 °C, 870–1470 °C and 1470–1700 °C, respectively [44,52]. Even traces of impurities inhibit kinetically and influence all phase transformations [44,52]. For instance, any additives such as alkali metal ions play a noteworthy role in forming a specific silica phase [44,50,51,52,53,54]. In a certain case, a type of doping ion affects the transition from amorphous silica to crystalline phases [52]. In particular, the size of the ion exhibits a certain relationship with the cell volume of the crystalline phase [52]. In that way, Na^+^ ion having the ionic radius of 1.02 Å favors the transition to cristobalite (cell volume = 171 Å^3^, density = 2.32 g cm^−3^, tetragonal) [52,61,62].

Mesoporous silica materials, SBA-15 were treated with different temperatures (500–650 °C) using Si/Na = 40. Although the peak intensity of the sample (SBA15-Na-40-600) treated at 600 °C was decreased, the mesostructure of both samples (SBA15-Na-40-500, SBA15-Na-40-600) treated at 500 and 600 °C were retained as shown in SAXS patterns of Figure 2A(a,b). With higher temperature of 650 °C, the mesostructure of SBA15-Na-40-650 was collapsed as shown in SXAS pattern of Figure 2A(c). The result can be due to the insertion of Na^+^ into silica matrix of pore walls. It is clearly supported by the XRD pattern with a small peak at 2θ = 21.9° indicating the production of cristobalite by the rearrangement of silica matrix. SBA15-Na-40-500 and SBA15-Na-40-600 did not produce crystalline silica species. The result means that it depends not only on the amount of alkali metal ion (Na^+^) but also on the temperature.

The morphologies of samples synthesized with different content (Si/metal ion = ∞–2.1) of Na^+^ at 700 °C were observed directly by SEM technique as shown in Figure 3. With the increase of Na^+^ content, the morphologies of the particles changed to spherical shapes with large pores in each particle. The collapse of the mesostructure in samples was observed directly by TEM technique (Figure 4). When the Na^+^ content was increased up to Si/Na = 2.1, the large pores were produced while the arrangement of mesochannels were collapsed. The results accord well with the XRD results in Figure 1A. These results can be explained by the rearrangement of silica matrix when the alkali metal ions are incorporated into silica by thermal treatment at elevated temperature.

The morphologies and the mesostructures of samples synthesized after thermal treatment with different temperatures (500–650 °C) using Si/Na = 40 were also observed directly by SEM and TEM techniques as displayed in Figure 5. Unlike SBA15-Na-40-500 and SBA15-Na-40-500 with the ordered mesostructures, the sample synthesized at 650 °C (SBA15-Na-40-650) has the collapsed mesostructure with the expanded mesopores. The result can be due to the insertion of Na^+^ into silica matrix of pore walls to produce crystalline silica (cristobalite). The results also correspond to the SAXS and XRD results in Figure 2.

Figure 6 shows the N_2_ adsorption/desorption isotherm curves (A) and the pore size distributions (B) of samples prepared with different content (Si/metal ion = ∞–2.1) of Na^+^ at 700 °C. As the increase up to Si/Na = 2.1, Pore diameter was increased 66.7 to 1779.9 Å while BET surface area and pore volume were decreased 713 to 11 m^2^g^−1^ and 0.92 to 0.03 cm^3^·g^−1^, respectively (Table 1). The increase in the pore diameter is due to the pore size expansion with the collapse of the mesostructures by Na^+^ incorporation into the silica matrix. The result accords well with the SEM and TEM results in Figure 4. Meanwhile, the decrease in surface area and pore volume was caused by the collapse of the ordered mesostructures with high surface area and large pore volume.

The physico-chemical properties of the samples synthesized after thermal treatment with different temperatures (500–650 °C) using Si/Na = 40 were obtained from N_2_ sorption results as shown in Figure 7. The sample synthesized at 650 °C (SBA15-Na-40-650) has the low surface area (25 m^2^ g^−1^) and small pore volume (0.04 cm^3^·g^−1^) compare to the SBA15-Na-40-500 and SBA15-Na-40-600 (Table 2). The pore diameter of SBA15-Na-40-650 with dual pores (119 Å and 364 Å) is larger size than that of SBA15-Na-40-500 (78.5 Å) and SBA15-Na-40-600 (66 Å) (Table 2). The results can be also contributed to the large pores from the rearrangement of silica matrix when the Na^+^ was inserted in the silica walls to produce cristobalite. The result shows clear that the change of the mesopores depends on the temperature in the presence of alkali metal ion (Na^+^). The results also accord well to the results of SEM and TEM in Figure 5.

Figure 8 shows XRD patterns at low angles (2θ = 1.2–6°) (A) and wide angle (10–80°) (B) of (a) SBA15-Li-40-700, (b) SBA15-K-40-700 and (c) SBA15-Ca-40-700 after thermal treatment at 700 °C using different metal ions (Li^+^, K^+^ and Ca^2+^) with Si/metal ion = 40. Interestingly, unlike SBA15-Li-40-700 and SBA15-K-40-700, SBA15-Ca-40-700 synthesized using Ca^2+^ as alkali earth cation source produced quartz species. It was reported that amorphous silica favored the formation of quartz (cell volume = 113 Å^3^, density = 2.65 g cm^−3^) with small cell volume after thermal treatment with small alkali metal ions (Li^+^) in size, while the formation of tridymite (cell volume = 2110 Å^3^, density = 2.27 g cm^−3^, monoclinic) with a large cell volume was favored when treated with large alkali metal ions (K^+^, Cs^+^) [52].

In our work, thermally treated SBA-15 at 700 °C using Ca^2+^ ion with higher valence and similar ion size (1.00 Å) compared to the Na^+^ (ionic radius of 1.02 Å) favored the transition to quartz, while thermally treated SBA-15 with Na^+^ exhibited the transition to cristobalite and quartz. The result can be due to the higher field strength of Ca^2+^ than that of Na^+^, based on the following Equation:Field strength of cation = *Z*/*r*2(1)
where *Z* is the valence and *r* is the ionic radius [26,63].

However, SBA15-Li-40-700 synthesized with Li^+^ (ionic radius of 0.76 Å) (Figure 8a) and SBA15-K-40-700 synthesized with K^+^ (ionic radius of 1.38 Å) (Figure 8b) did not produce crystalline silica species under the condition in this work. It may be assumed that the SBA-15 with crystalline structure may be prepared even in the presence of Li^+^ and K^+^ if the thermal treatment temperature is higher than 700 °C according to previous works [45,51,52]. As described in the Introduction, a few research groups obtained silica materials with crystalline structure when they used such high thermal treatment temperature as >800 °C. In the present work, we intended to study the effect of Li^+^ and K^+^ on the wall matrix structure of mesoporous silicas under various reaction conditions (different concentration of Li^+^ and K^+^, different annealing temperature and annealing time, etc.) through in-depth studies, even though the SBA-15 with crystalline walls was not obtained.

The morphologies and the mesostructures of SBA15-Li-40-700, SBA15-K-40-700 and SBA15-Ca-40-700 synthesized by thermal treatment at 700 °C using different metal ions (Li^+^, K^+^ and Ca^2+^) with Si/metal ion = 40 were observed directly using SEM and TEM techniques (Figure 9). SBA15-Ca-40-700 showed larger pore size (802.5 Å) than that of SBA15-Li-40-700 (79.6 Å) and SBA15-K-40-700 (74.7 Å) by the rearrangement of silica matrix due to the incorporation of Ca^2+^ inducing crystallization (Figure 9f, Figure 10 and Table 3). On the other hand, surface area (35 m^2^·g^−1^) and pore volume (0.27 cm^3^·g^−1^) of SBA15-Ca-40-700 were lower values than that of SBA15-Li-40-700 and SBA15-K-40-700 (Table 3). The result can be caused by the larger pore size and the disordered pores of SBA15-Ca-40-700 compare to SBA15-Li-40-700 and SBA15-K-40-700 with the ordered mesopores.

## 5. Conclusions

In this work, we synthesized mesoporous silica with the crystalline wall via thermal treatment with SBA-15 in the presence of alkali or alkali earth metal ions (Na^+^, Li^+^, K^+^ and Ca^2+^) as catalyst. With Na^+^ (ionic radius of 1.02 Å) as source of alkali metal ion, mesoporous silica SBA-15 showed clearly the production of crystalline silica walls (i.e., quartz and cristobalite) when the Na^+^ content of Si/Na = 20 or more was used at the temperature of 650 °C or higher. When the Na^+^ amount was increased up to Si/Na = 2.1 at 700 °C (SBA15-Na-2.1-700), the crystalline silica was dominated by cristobalite. Meanwhile, pore diameter was increased with dual size (18.3 and 972.7 Å) compared to the pristine SBA-15, but the surface area and pore volume were decreased to 11 m^2^·g^−1^ and 0.03 cm^3^·g^−1^, respectively. After thermal treatment SBA-15 with Si/Ca = 40 at 700 °C (SBA15-Ca-40-700), the mesoporous silica showed clearly the production of crystalline silica wall dominated by quartz. However, SBA15-Li-40-700 prepared with Li^+^ (ionic radius of 0.76 Å) and SBA15-K-40-700 prepared with K^+^ (ionic radius of 1.38 Å) did not produce crystalline silica species under the same condition. Pore size, surface area and pore volume of SBA15-Ca-40-700 were 802.5 Å, 35 m^2^·g^−1^ and 0.27 cm^3^·g^−1^, respectively. Transition of amorphous silica to crystalline structure can be explained by the rearranging silica matrix when alkali or alkali earth metal ions are inserted into silica matrix at elevated temperature. Na^+^ ion in silica was favorable for the formation of cristobalite structure with large cell volume (cell volume = 171 Å^3^). On the other hand, Ca^2+^ ion with the lager valence (i.e., the higher field strength) than Na^+^ ion in silica was favorable for the formation of quartz structure with smaller cell volume (cell volume = 113 Å^3^) than cristobalite.

## Data Availability

Data is contained within the article.

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
