# Peer review of "SBA-15 with Crystalline Walls Produced via Thermal Treatment with the Alkali and Alkali Earth Metal Ions"

_materials, 2021, doi:10.3390/ma14185270_

Round 1

Reviewer 1 Report

Mesoporous nanoparticles are important nanomaterials for research and application. The author in this work applied the thermal treatment on SBA-15 in the presence of alkali or alkali earth metal ions (Na+ , Li+ , K+ , and Ca2+) to synthesize mesoporous silica with the crystalline wall. The conditions including ratio between metal ion and silicon, sintering temperature have been studied to synthesize the crystalline walls in mesoporous silica, such as Si/Na=20, or Si/Ca=40 at 650 C and 700C respectively. The phases have been identified via small angle, low/wide angle XRD technique. The pore sizes and surface area have been studied via BET. Transition of amorphous silica to crystalline structure can be due to the rearrangement of silica matrix by insertion of alkali or alkali earth metal ions into silica matrix at high temperature. Na+ ion in silica was favorable for the formation of cristobalite structure with large cell volume (cell volume=171 Å 3 ). On the other hand, Ca2+ ion with the lager valence (i.e. the higher field strength) than Na+ ion in silica was favorable for the formation of quartz structure with smaller cell volume (cell volume=113 Å 3 ) than cristobalite. This study is much interesting to a broad of readers and thus would be recommended for the acceptance in Materials. The questions are shown below,

  • How do the authors explain the amorphous structure of SBA-15 using Li+, and K+ ? Is it because the authors did not reach the ideal conditions or only Na+ and Ca+ could create the crystalline walls.
  • The authors argue crystalline structure is due to the rearrangement of silica matrix. How did the authors characterize the crystalline structure, such as TEM images. The current TEM images focus on mesopores.

Author Response

Responses to reviewers’ comments

To reviewer 1

Mesoporous nanoparticles are important nanomaterials for research and application. The author in this work applied the thermal treatment on SBA-15 in the presence of alkali or alkali earth metal ions (Na+ , Li+ , K+ , and Ca2+) to synthesize mesoporous silica with the crystalline wall. The conditions including ratio between metal ion and silicon, sintering temperature have been studied to synthesize the crystalline walls in mesoporous silica, such as Si/Na=20, or Si/Ca=40 at 650 C and 700C respectively. The phases have been identified via small angle, low/wide angle XRD technique. The pore sizes and surface area have been studied via BET. Transition of amorphous silica to crystalline structure can be due to the rearrangement of silica matrix by insertion of alkali or alkali earth metal ions into silica matrix at high temperature. Na+ ion in silica was favorable for the formation of cristobalite structure with large cell volume (cell volume=171 Å 3 ). On the other hand, Ca2+ ion with the lager valence (i.e. the higher field strength) than Na+ ion in silica was favorable for the formation of quartz structure with smaller cell volume (cell volume=113 Å 3 ) than cristobalite. This study is much interesting to a broad of readers and thus would be recommended for the acceptance in Materials. The questions are shown below;

Q.1. How do the authors explain the amorphous structure of SBA-15 using Li+, and K+? Is it because the authors did not reach the ideal conditions or only Na+ and Ca2+ could create the crystalline walls.

Answer) Thank you for your valuable comment. As mentioned in the 'Introduction' section, the silica materials with crystalline structure in the presence of Li+ and K+ were reported by several research groups (Venezia et al., J. Solid State Chem. 2001, 161, 373-378; Shinohara et al., Ind. Health 2004, 42, 277-285; Deepak et al., Mater. Res. 2004, 19, 2216-2221). Those previous works used different experimental conditions in comparison to ours. They prepared their materials at higher temperature (above 800 oC) than our condition (i.e. below 700 oC). In our study, the crystalline silica was produced in the presence of Na+ and Ca2+, while the crystalline silica was not formed in the presence of Li+ and K+ (heat treatment temperature below 700 oC and Si/Li+ (or K+) = 40). We intended to study the effect of Li+ and K+ on the wall matrix structure of mesoporous silicas under various reaction conditions (different concentration of Li+ and K+, different annealing temperature and annealing time, etc.) through an in-depth studies.

We newly mentioned this point in the revised version in Page 12, L387- Page 13, L396.

Q.2. The authors argue crystalline structure is due to the rearrangement of silica matrix. How did the authors characterize the crystalline structure, such as TEM images. The current TEM images focus on mesopores.

Answer) Thank you for your valuable comment. Unfortunately, we could not obtain TEM images of crystalline silica in this work due to the limitation of instruments in our university. However, the XRD patterns clearly showed the production of crystalline silica (cristobalite and quartz). Most of previous studies also used XRD patterns to prove the production of crystalline silica well. Therefore, we consider that the XRD results are enough for explaining our research results. However, we also mentioned the necessity of TEM measurements (with electron diffraction) for more accurate characterization of the crystalline wall structures in Page 11, L290- 295.

May we ask your warm understanding?

We thank you very much for your valuable comments and suggestions. Our revision is highlighted in red in this revised manuscript. We did our best to incorporate your valuable comments in this revised manuscript. We believe the quality of this revised manuscript has been significantly improved. We hope that our revision could have been done successfully.

Reviewer 2 Report

This is an interesting work. I think that it can be published after considering the following comments.

  1. The crystalline mesopore wall should be further demonstrated by other techniques.
  2. Why was no micropore distribution detected by adsorption technique?

Author Response

To reviewer 2

This is an interesting work. I think that it can be published after considering the following comments.

Q.1. The crystalline mesopore wall should be further demonstrated by other techniques.

Answer) Thank you for your valuable comment. In our study, the XRD technique was mainly used to describe the crystalline silica structure. The results of XRD clearly showed the production of crystalline silica (cristobalite and quartz). Most of previous studies also used the results of XRD patterns to prove the production of crystalline silica well. Therefore, we consider that the XRD results are enough for explaining our research results. Sometimes, high resolution TEM has been used to show the crystalline structure but could not be done for this work due to the instrumental limitation in our university. However, we also mentioned the necessity of TEM measurements (with electron diffraction) for more accurate characterization of the crystalline wall structures in Page 11, L290- 295.

Q.2. Why was no micropore distribution detected by adsorption technique?

Answer) Thank you for your valuable comment. We used an instrument that was optimized with high reliability to mainly measure pores larger than mesopore size (above 1.5 nm). Therefore, information on micropores could not be measured for our work and thus not mentioned in this study. In this work, we focused on the changes in mesopore structure and silica matrix structure according to the concentration of various alkali ions in mesoporous silica and the annealing temperature.

May we ask your warm understanding?

We thank you very much for your valuable comments and suggestions. Our revision is highlighted in red in this revised manuscript. We did our best to incorporate your valuable comments in this revised manuscript. We believe the quality of this revised manuscript has been significantly improved. We hope that our revision could have been done successfully.

Reviewer 3 Report

The manuscript presents the studies of the amorphous wall’s crystallization in SBA-15 in the presence of alkali metal ions. The manuscript has to be significantly corrected and improved:

  • English must be corrected – starting from the manuscript title.
  • The manuscript is written like rapport, not scientific paper. Apart from presentation of the results also their analysis and discussion are necessary.
  • Lines 88-89: “… Then, it was allowed to stand for crystallization under static hydrothermal conditions at 100°C for 24 h in a teflon bottle. The crystallized product was filtered, …”. The obtained SBA-15 material is composed of amorphous silica. The crystallization of such amorphous silica was conducted in the next step. This part must be corrected.
  • Lines 102-104: “… The materials synthesized were called as SBA15-A-C-T, where A represents metal ion species, C represents the ratio of Si/Na, T is the heating temperature …”. This is not valid for other than sodium cations. It would be better to modify this part in the following way “… where A represents metal ion species, C represents the ratio of Si/A …”.
  • Are authors sure that “the large pores”, observed for the alkali treated SBA-15 samples, are not interparticle spaces? Please, verify this issue.
  • The physico-chemical properties of the SBA-15 samples modified with lithium, potassium and calcium, similarly to the samples modified with sodium, should be compared in table.

Author Response

To reviewer 3

The manuscript presents the studies of the amorphous wall’s crystallization in SBA-15 in the presence of alkali metal ions. The manuscript has to be significantly corrected and improved:

Q.1. English must be corrected – starting from the manuscript title.

Answer) By your advice, we have smoothed out the English sentences of the manuscript throughout the manuscript.

In this context, we also changed title of the manuscript ‘Mesoporous Silica with Crystalline Mesopore Walls by Thermal Treatment with Alkali Metal Ions’ to ‘Mesoporous Silica with Crystalline Walls Produced via Thermal Treatment with the Alkali and Alkali Earth Metal Ions’ for clear meaning.

Q.2. The manuscript is written like report, not scientific paper. Apart from presentation of the results also their analysis and discussion are necessary.

Answer) According to your advice, we have separated the 'Results and Discussion' section into a '3. Results' section (Page 3, L117 – Page 10, L285) and a '4. Discussion' section (Page 10, L286 – Page 13, L408) in the manuscript to highlight more about the Discussion.

Q.3. Lines 88-89: “… Then, it was allowed to stand for crystallization under static hydrothermal conditions at 100°C for 24 h in a teflon bottle. The crystallized product was filtered, …”. The obtained SBA-15 material is composed of amorphous silica. The crystallization of such amorphous silica was conducted in the next step. This part must be corrected.

Answer) Thank you for your valuable advice. We changed ‘Then, it was allowed to stand for crystallization under static hydrothermal conditions at 100 oC for 24 h in a teflon bottle.’ to ‘Then, it was allowed to stand under static hydrothermal conditions at 100 oC for 24 h in a Teflon bottle.’. (That is, ‘for crystallization’ was deleted) (Page 2, L88-90)

Q.4. Lines 102-104: “… The materials synthesized were called as SBA15-A-C-T, where A represents metal ion species, C represents the ratio of Si/Na, T is the heating temperature …”. This is not valid for other than sodium cations. It would be better to modify this part in the following way “… where A represents metal ion species, C represents the ratio of Si/A …”.

Answer) Thank you for your kind suggestion. We changed ‘Si/Na’ to ‘Si/A’. (Page 2, L101)

Q.5. Are authors sure that “the large pores”, observed for the alkali treated SBA-15 samples, are not interparticle spaces? Please, verify this issue.

The physico-chemical properties of the SBA-15 samples modified with lithium, potassium and calcium, similarly to the samples modified with sodium, should be compared in table.

Answer) As shown in SEM images {Figure 3(e)~3(h), Figure 5(c)}, it is clear that large pores were produced in each particle as rearrangement of silica matrix at high temperature in the presence of the metal ions. (The SEM image obtained at low magnification in Figure 9c does not show the pore images.) And the large pores are more clearly observed in enlarged TEM images (Figure 4(c)~4(h), Figure 5(c), Figure 9(f)). Therefore, based on these results, it is clear that the large pores were not generated by interparticles, but were formed during the rearrangement of the silica matrix. It is already well described in Page 5,L175- Page 6, L199.

By your advice, we added as ‘Table 3’ in the manuscript the summarized physico-chemical properties of the SBA-15 samples modified with lithium, potassium and calcium (Page 10).

We thank you very much for your valuable comments and suggestions. Our revision is highlighted in red in this revised manuscript. We did our best to incorporate your valuable comments in this revised manuscript. We believe the quality of this revised manuscript has been significantly improved. We hope that our revision could have been done successfully.

Round 2

Reviewer 3 Report

I suggest the acceptance of the manuscript in the present form/